# Mininvasive Cytoreduction Surgery plus HIPEC for Epithelial Ovarian Cancer: A Systematic Review

**DOI:** 10.3390/medicina59030421

**Published:** 2023-02-21

**Authors:** Carlo Ronsini, Francesca Pasanisi, Pierfrancesco Greco, Luigi Cobellis, Pasquale De Franciscis, Stefano Cianci

**Affiliations:** 1Department of Woman, Child and General and Specialized Surgery, University of Campania Luigi Vanvitelli, 80138 Naples, Italy; 2Unit of Gynecologic Oncology, Department of Woman, Child and Public Health, A. Gemelli, IRCCS, University Hospital Foundation, 00198 Rome, Italy; 3Unit of Gynecology and Obstetrics, Department of Human Pathology of Adult and Childhood “G. Barresi”, University of Messina, 98122 Messina, Italy

**Keywords:** Cisplatin in Hyperthermic Intraperitoneal Chemotherapy (HIPEC), ovarian cancer, minimally invasive surgery

## Abstract

*Background and objectives:* The Gold-Standard treatment for Advanced Epithelial Ovarian Cancer remains cytoreductive surgery followed by systemic chemotherapy. Surgery can be performed either by an open or minimally invasive approach (MIS), although the former remains the most widely used approach. Recently, Van Driel et al. proved that adding 100 mg/m^2^ of Cisplatin in Hyperthermic Intraperitoneal Chemotherapy (HIPEC) at Interval Debulking Surgery (IDS) gives a disease-free survival (DFS) advantage. Similarly, Gueli-Alletti et al. demonstrated how the MIS approach is feasible and safe in IDS. Moreover, Petrillo et al. reported pharmacokinetic profiles with a higher chemotherapy concentration in patients undergoing HIPEC after MIS compared with the open approach. Therefore, the following review investigates the oncological and clinical safety consequences of the association between MIS and HIPEC. *Methods:* Following the recommendations in the Preferred Reporting Items for Systematic Reviews and Meta-Analyses (PRISMA) statement, we systematically searched the PubMed and Scopus databases in April 2022. Studies containing data about oncological and safety outcomes were included. We registered the Review to the PROSPERO site for meta-analysis with protocol number CRD42022329503. *Results:* Five studies fulfilled inclusion criteria. 42 patients were included in the review from three different Gynecological Oncological referral centers. The systematic review highlighted a Recurrence Rate ranging between 0 and 100%, with a 3-year Platinum-Free Survival between 10 and 70%. The most common HIPEC drug was Cisplatin, used at concentrations between 75 and 100 mg/m^2^ and at an average temperature of 42 °C, for 60 to 90 min. Only 1 Acute Kidney Insufficiency has been reported. *Conclusions:* The scarcity of clinical trials focusing on a direct comparison between MIS and the open approach followed by HIPEC in EOC treatment does not make it possible to identify an oncological advantage between these two techniques. However, the safety profiles shown are highly reassuring.

## 1. Introduction

The Gold-Standard treatment for Advanced Epithelial Ovarian Cancer (AEOC) remains cytoreductive surgery and systemic chemotherapy [1]. The succession between these two depends on the extent of the disease at diagnosis, the intraoperative risk of complications, and the patient’s health status. Therefore, two different treatment schemes are accepted. On the one hand, primary cytoreductive surgery (PDS) is followed by adjuvant platinum-based therapy when complete cytoreduction (CC-0) is expected. Alternatively, neoadjuvant chemotherapy, followed by interval cytoreductive surgery (IDS) and completion with adjuvant chemotherapy, is used whenever PDS may be burdened with a high risk of complications or complete cytoreduction cannot be achieved (CC-0) [2].

In 2020, Fagotti et al. demonstrated how the two treatment regimens are equal in oncologic outcomes, whereas PDS is more likely to be associated with postoperative complications [3]. Although open surgery remains the most practiced surgical approach even in IDS, there is evidence that, in selected cases, a minimally invasive (MIS) approach is also possible [4]. Additionally, in 2018, van Driel et al. proved how the addition of 100 mg/m^2^ of Cisplatin in Hyperthermic Intraperitoneal Chemotherapy (HIPEC) at IDS gives an advantage in disease-free survival (DFS) [5]. Another reference center confirmed this finding [6].

The trial from Van Driel represented an initial standardization of HIPEC treatment, which historically could differ in the drug used, dosage, and duration of administration. Finally, the mode of administration may also affect the pharmacokinetic profile of HIPEC, depending on the higher pressures that can be achieved laparoscopically (LPS) [7,8]. On the other hand, the role that HIPEC may have in different types of patients with Epithelial Ovarian Cancer (EOC) remains debated. A phase III trial is currently underway to investigate the potential of the association of HIPEC with PDS [9]. Instead, the role of HIPEC administration in ovarian cancer relapses (ROC) is much more controversial [10,11,12]. For these reasons, it is not easy to delineate the weight of HIPEC in the treatment of EOC. In this scenario, patients undergoing laparoscopic cytoreductive surgery may represent highly selected patients with little tumor burden in whom the addition of HIPEC can be free of many other confounders. Consequently, we attempted to unravel some of the fog by focusing our systematic review on the possibility of combining HIPEC with minimally invasive approaches in EOC.

## 2. Materials and Methods

The methods for this study were specified a priori based on the recommendations in the Preferred Reporting Items for Systematic Reviews and Meta-Analyses (PRISMA) statement [13]. We registered the Review to the PROSPERO site for meta-analysis with protocol number CRD42022329503.

### 2.1. Search Method

We performed a systematic search for articles about the association of HIPEC and minimal invasive cytoreductive surgery and EOC in the PubMed Database and the Scopus Database in April 2022. We implemented no restriction on the publication year or the country. We considered only studies published entirely in English. Search inputs were “(((Minimal*) OR (Laparoscopic)) AND ((Hipec) OR (H.I.P.E.C) OR (Hyperthermic Intraperitoneal Chemotherapy)) AND ((Ovarian) OR (ovary))) Filters: English” and “(TITLE-ABS-KEY (hyperthermic AND intraperitoneal AND chemotherapy) OR TITLE-ABS-KEY (hipec) OR TITLE-ABS-KEY (h.i.p.e.c.) AND ALL (ovarian) AND NOT TITLE-ABS-KEY (colo*) AND NOT TITLE-ABS-KEY (pseudo*) AND NOT TITLE-ABS-KEY (pancreas*) AND NOT TITLE-ABS-KEY (gastr*) AND NOT TITLE-ABS-KEY (mesot*) AND (LIMIT-TO (PUBSTAGE, “final”)) AND (LIMIT-TO (LANGUAGE, “English”))” respectively for the PubMed and Scopus databases.

### 2.2. Study Selection

Study selection was made independently by F.P. and P.G. In case of discrepancy, C.R. decided on inclusion or exclusion. Inclusion criteria were: (1) studies that included patients with EOC, at any stage or relapsed, who underwent laparoscopic cytoreductive surgery plus HIPEC; (2) studies that reported at least one outcome of interest (Platinum Free Survival (PFS); Overall Survival (OS); Recurrence Rate (RR), Complete Cytoreduction (CC-0) rate; Early complication; Late complication; Acute Kidney Insufficiency (AKI); (3) peer-reviewed articles, published originally. Non-original studies, preclinical trials, animal trials, abstract-only publications, and articles in a language other than English were excluded. If possible, the authors of studies that were only published as congress abstracts were tried to be contacted via email and asked to provide their data. The studies selected and all reasons for exclusion are mentioned in the Preferred Reporting Items for Systematic Reviews and Meta–Analyses (PRISMA) flowchart (Figure 1). All included studies were assessed regarding potential conflicts of interest.

### 2.3. Data Extraction

F.P. and D.A. extracted data for all relevant series and case reports. They extracted data on tumor characteristics (stage, histological subtype, LVSI status, grading), surgical approach, morbidity and oncological issues such as Recurrences, Deaths, Recurrence Rate (RR) and Complete Cytoreduction (CC-0) rate. We also collected data about HIPEC drugs and administration regimes (Time of exposure, Concentration, Temperature). Data about adjuvant treatment were collected (number of cycles, drugs). Additionally, data about complications in the first 30 days after surgery (Early) and past this time (Late) were extracted and graded according to the Clavien-Dindo scale [14].

### 2.4. Quality Assessment

The quality of the included studies was assessed using the Newcastle–Ottawa scale (NOS) [15]. This assessment scale uses three broad factors (selection, comparability, and exposure), with the scores ranging from 0 (lowest quality) to 8 (best quality). Two authors (D.A. and F.P.) independently rated the quality of the studies. Any disagreement was subsequently resolved by discussion or consultation with P.D.F. We reported the NOS Scale in Appendix A.

## 3. Results

### 3.1. Studies’ Characteristics

After the database search, 1023 articles matched the search criteria. After removing records with no full-text, duplicates, and wrong study designs (e.g., reviews), 176 were suitable for eligibility. Of those, five matched inclusion criteria and were included in the systematic review. Two of them were non-comparative, single-armed studies evaluating the effect of HIPEC in ovarian cancer. We extracted data about patients treated by the MIS approach. The other three were comparative studies between open and MIS approaches for ovarian cancer, followed by HIPEC (Figure 1). The countries where the studies were conducted, the publication year range, the design of the studies, the characteristics of the population, surgical treatment, HIPEC drugs, adjuvant treatment, mean Follow Up (FUP), and the number of participants is summarized in Table 1 [7,8,16,17,18]. The quality of all studies was assessed by the NOS [15] (Appendix A). Overall, the publication years ranged from 2005 to 2019. In total, 42 patients who underwent MIS followed by HIPEC were enrolled. Follow up ranged from 7 up to 36 months on average.

### 3.2. Outcomes

A total of 42 patients were included in the review. Four of the five selected studies presented at least one oncological data of interest, such as residual tumor after surgery (CC-0), recurrence rate (RR), Platinum-Free Survival (PFS). None of the included studies presented OS data. Four of the five selected studies presented at least one security data of interest, such as early complication after surgery, late complication after surgery, and Acute Kidney Insufficiency (AKI) rate. It should be noted that papers by Fagotti, 2013 [16] and 2014 [8] represent the same patient population placed in the context of a study focused on oncological outcomes and the other on safety outcomes. By the publication date in 2013, Fagotti et al. [16] firstly reported data about performing the HIPEC and MIS approach. They focused their retrospective case series on isolated platinum-sensitive ovarian cancer recurrences, showing optimal surgical feasibility (CC-0 obtained in 100% of the cohort) and a 3 year PFS of 10% in a Follow-Up (FUP) period going from the procedure to relapse (6–37 months). The administered drug during HIPEC was Cisplatin at a concentration of 75 mg/m^2^, at a temperature of 41.5° for 60 min. The following year, the same group [8] presented data about feasibility, showing no Clavien-Dindo [14] grade 3 complications related to the procedure. This second paper considered a group treated for the same disease with an open approach as a control group. This group also showed no complications, but the mean hospitalization stay was twice that of the MIS group (8 days vs. 4 days, *p* = 0.002). In 2018, Arjona-Sanchez et al. [17] reported their experience treating peritoneal carcinomatosis with the MIS approach followed by HIPEC. Out of the 162 patients, two were due to EOC, stage IIIC. Both the patients underwent MIS cytoreductive surgery at IDS after four cycles of Carboplatin-Paclitaxel, plus HIPEC with Paclitaxel 120 mg/m^2^. No data about temperature or the duration of treatment were reported. Both reached CC-0, and no recurrence was reported after 6 and 9 months of FUP. In 2018, Petrillo et al. [7] analyzed pharmacokinetic characteristics of patients treated with Cisplatin at a concentration of 75 mg/m^2^, at a temperature of 41.5° for 60 min at the time of cytoreductive surgery, comparing MIS to the open approach. It represented a second paper from the group of Fagotti et al. [8,16], who are previously discussed. The primary endpoint was pharmacokinetic characteristics, with a higher concentration of plasmatic Cisplatin after 120 min from HIPEC in the MIS group (0.511 picomol/L in MIS approach vs. 0.254 picomol/L in open approach; *p* = 0.012). They reported clinical data from this series. They obtained CC-0 in the whole series, a RR of 33.3%, and a 3 year PFS of 70% (vs. 35% of open surgery; *p* = 0.054 in 36 months of FUP. Only 1 AKI was reported in this series (11.1%). Lastly, in 2020, Morton et al. [18] published a retrospective comparison between MIS and the open approach followed by HIPEC with Cisplatin at a concentration ranging between 80 mg/m^2^ and 100 mg/m^2^, at a temperature ranging between 41° and 43° for 90 min, at IDS after four cycles of Carboplatin-Paclitaxel. They treated 10 Patients with the MIS approach (8 with single-port laparoscopy, 1 with standard laparoscopy, 1 with robotic surgery). They reported a conversion rate to classic laparotomy of 4%, with a CC-0 of 70%, which is lower but not a statistically significant difference from the open approach group (77.5%; *p* = 0.39). A 20% Clavien-Dindo [13] grade 3 complication was reported in the MIS group (vs 27.5% in the open approach group; *p* = 0.64). 3-years PFS was 15%.

Overall, all of the studies apart from the one by Arjona-Sanchez [17] used Cisplatin as HIPEC drugs, with concentrations ranging between 75 and 100 mg/m^2^, at temperatures never lower than 41° and never higher than 43°, for 60 to 90 min. All of the series apart from the one by Morton [18] obtained CC-0 in all patients. RR ranged between 0 and 100%, with a 3-year PFS between 10 and 70%. Oncological results are summarized in Table 2. The complication ≥ G3 complication rate in the first 30 days after surgery was between 0 and 30%. No complications after the 30 days were reported. Only 1 AKI was reported. Safety results are summarized in Table 3.

## 4. Discussion

Ovarian carcinoma remains a disease with a high recurrence rate [19]. Therefore, any strategy that can optimize its chronicity must be sought. HIPEC is a helpful ally [5,9]. In the same philosophy, surgery-related morbidity and its repetition must be minimized to improve replicability without affecting the quality of life. Therefore, the modern trend to convert surgery to a minimally invasive approach may also extend to EOC treatment. Consequently, it has become necessary to intercept the possibility of combining MIS with HIPEC.

The feasibility of laparoscopic peritonectomy followed by HIPEC was first demonstrated in animal models [20]. Several series showed the feasibility of the MIS approach in peritoneal carcinosis. Firstly, Esquivel [21], in 13 patients with low peritoneal tumor volume, and no small bowel involvement from gastrointestinal cancer, attempted cytoreductive surgery followed by HIPEC, reported a laparotomy conversion rate of 20%, and grade 3/4 complications in only two patients. Next, Passot et al. [22] and Fish et al. [23] reported a comparison of laparoscopic vs. laparotomic cytoreduction followed by HIPEC, respectively, in 16 patients affected by peritoneal surface malignancies and 17 low-grade appendiceal mucinous neoplasms with no peritoneal dissemination. Both studies demonstrated a shorter period of hospitalization and a lower complication rate in the MIS arm, but both studies were limited by a low peritoneal dissemination of disease. This element remains to define the choice of the surgical approach. In fact, in all reported series, surgical complexity was minimal and reserved for highly selected cases. Only the study by Arjona-Sanchez [17] reported diaphragmatic peritonectomy as a procedure. All other studies reported low complexity surgery often limited to hysterectomy [24], Lymphadenectomy [25] and omentectomy in cases of first diagnosis or adhesiolysis and single excision for recurrences. Therefore, the high selection of patients may hide the proper role exerted by HIPEC at the time of MIS, since part of the oncological outcomes can be owed to the low surgical complexity more than to the procedure [26]. However, on the contrary, this review shows us the high safety profile of the association between HIPEC and MIS. Complication rates were close to zero and, in any case, lower than those reported in the scientific literature.

Moreover, we have previously demonstrated that the addition of HIPEC at the time of debulking surgery does not affect the quality of life of patients [19]. Finally, following the previously stated principle of the chronicization of treatments, it is necessary to consider the timing of the use of HIPEC. The reported case histories range from PDS surgery to secondary and tertiary surgery. It is, therefore, difficult to identify the best surgery to associate HIPEC with. At present, its efficacy in IDS [5] is proven, and its function in PDS [9] and in platinum-sensitive recurrences is under investigation (HORSE Trial, NCT01539785). However, it has been previously demonstrated by Cianci et al. [12] that repeat HIPEC is feasible in platinum-sensitive recurrences, opening up scenarios for the routine application of HIPEC in EOC surgery. Another interesting finding of this review is the exceptionally high PFS (70%) shown by patients undergoing MIS followed by HIPEC in Petrillo et al. [7]. In particular, this favorable outcome correlates with the highest plasmatic, tissue, and urinary concentrations of Cisplatin recorded. The authors explained this finding to be the consequence of the higher pressures reached during laparoscopic infusion. HIPEC is to be understood as a dynamic process influenced by multiple variables related to the characteristics of the drug (viscosity, molecular weight, polarity) and the mode of administration (pressure, temperature, size of the tumor tissue). These characteristics condition its absorption by the patient and affect its efficacy and safety, especially at the level of renal function [27]. Therefore, it is not surprising that the only case of AKI occurred in these patients.

Ultimately, our review is severely limited by several factors: the small sample number, the almost total number of patients attributed to the same working group [8,16] the paucity of comparative works, and the abundance of retrospective series. This is related to the extreme rarity of the investigated condition, which represents a peculiarity within a rare pathology such as EOC. Instead, its strength lies in the research’s rigor, since an in-depth analysis of the subject throughout the present literature has been run out. Finally, although the designs of the studies do not express themselves in terms of oncological advantage over an open approach, they show data of high tolerance that do not preclude the possibility of associating HIPEC to MIS surgery, which seems to be reserved for highly selected cases of patients, in which the MIS approach still allows the achievement of CC-0.

## 5. Conclusions

The role of HIPEC in the treatment of advanced EOC is still debated, due to the lack of consistent evidence. There need to be more clinical trials focusing on a direct comparison between MIS and the open approach followed by HIPEC in EOC treatment to identify an oncological advantage between these two techniques. However, the safety profiles shown are highly reassuring. This data allows us to lay the foundation for future work and does not preclude the MIS approach, where applicable, to patients who are candidates for HIPEC.

## Figures and Tables

**Figure 1 medicina-59-00421-f001:**
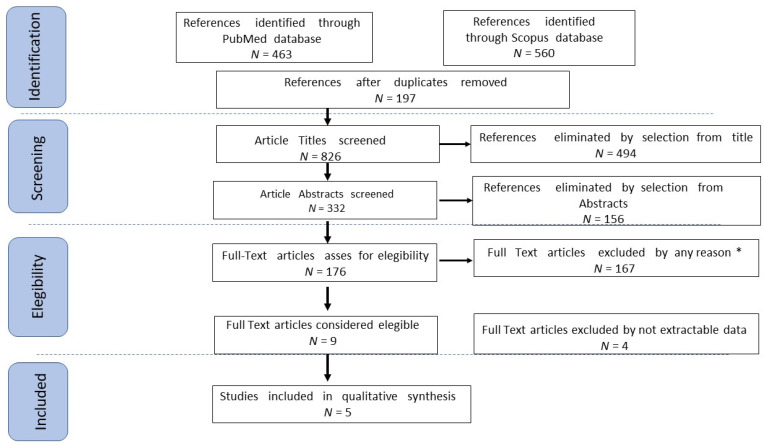
Preferred Reporting Items for Systematic Reviews and Meta–Analyses (PRISMA) flowchart.

**Table 1 medicina-59-00421-t001:** Studies characteristics.

Single Arm Studies									
Name	Country	Study Design	Study Year	FIGO Stage/Population	Surgical Treatment	HIPEC Drugs	Adjuvant Treatment	N of Participants	Mean FUP Months
Fagotti, 2013 [16]	Italy	Retrospective Observational Monocentric study	2009–2013	Isolated platinum sensitive recurrence	SCS	Cisplatin 75 mg/m^2^; 41.5 °C, 60 min	6 Carboplatin–Paclitaxel	10	10
Arjona-Sanchez, 2018 [17]	USA	Prospective Cohort Monocentric Study	2016–2018	IIIC	IDS	Paclitaxel 120 mg/m^2^;	Neoadjuvant 4 Carboplatin–Paclitaxel; 2 Carboplatin–Paclitaxel	2 ^	7, 5
**Comparative Studies**									
Fagotti, 2014 [8]	Italy	Retrospective Cohort Monocentric Study	2005–2013	Isolated platinum sensitive recurrence	SCS	Cisplatin 75 mg/m^2^; 41.5 °C, 60 min	6 Carboplatin–Paclitaxel	11 ^	–
Petrillo, 2019 [7]	Italy	Prospective Cohort Monocentric Study	2013–2016	Platinum sensitive recurrence	SCS	Cisplatin 75 mg/m^2^; 41.5 °C, 60 min	6 Carboplatin–Paclitaxel	9 ^	36
Morton, 2020 [18]	USA	Retrospective cohort Monocentric Study	2017–2019	III, IV	IDS	Cisplatin 80/100 mg/m^2^; 41–43 °C; 90 min	Neoadjuvant 3/4 Carboplatin–Paclitaxel; 2 Carboplatin–Paclitaxel	10 ^	15, 1

HIPEC: Hyperthermic Intraperitoneal Chemotherapy; FIGO: International Federation of Gynecology and Obstetrics; PDS: Primary Debulking Surgery; IDS: Interval Debulking Surgery; SCS: Secondary Cytoreductive Surgery; FUP: Follow Up; min: minutes; m: meters; °C: Celsius; ^: Sub-analysis of the entire cohort.

**Table 2 medicina-59-00421-t002:** Oncological Outcomes.

Name	CC-0 (%)	RR (%)	3Y-PFS (%)	3Y-OS (%)
Fagotti, 2013 [16]	100	100	10	–
Arjona-Sanchez, 2018 [17]	100	0	–	–
Perillo et al., 2019 [7]	100	33.3	70	–
Morton et al., 2020 [18]	70	–	15	–

CC: Complete cytoreduction; RR: Recurrence Rate; 3Y-PFS: 3 Years-Platinum Free Survival; 3Y-OS: 3 Years-Overall Survival.

**Table 3 medicina-59-00421-t003:** Safety Outcomes.

Name	Early Complication > G3 * (%)	Late Complication > G3 ** (%)	AKI ^ (%)
Fagotti, 2014 [8]	0	0	0
Arjona-Sanchez, 2018 [17]	0	–	0
Petrillo, 2019 [7]	11.1	0	11.1
Morton, 2020 [18]	20	–	0

* In the first 30 days after surgery, according to Clavien-Dindo Classification; ** Over 30 days after surgery, according to Clavien-Dindo Classification; ^ Acute Kidney Insufficiency.

## Data Availability

Not applicable.

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
