# Peer review of "Mininvasive Cytoreduction Surgery plus HIPEC for Epithelial Ovarian Cancer: A Systematic Review"

_medicina, 2023, doi:10.3390/medicina59030421_

Round 1

Reviewer 1 Report

The topic of the work is very interesting "Mininvasive Cytoreduction Surgery plus HIPEC for Epithelial 2 Ovarian Cancer: a Systematic Review", but it would seem that in the period from 2005 to 2019, only 5 articles presented research on this topic.

 My questions:

Apart from these 5 papers (originating from Italy and the USA), have you encountered this type of research in other countries?

Has this topic never been addressed before and never since?

 Minor notes:

1. Please note the values of 75 mg/m2 and 100 mg/m2. (Sometimes there are spaces and superscripts, sometimes not.)

2. There are many spaces in section 2.1.

3. Captions under Table 1 and Table 2 are hard to read (too many spaces).

4. In subsection 2.3 it says "FP e DA extracted" and shouldn't it say "FP and DA extracted"?

Author Response

Dear Reviewer,

Thank You for taking the time to review our manuscript and for your comments. They are crucial and valuable to us in raising the quality standard of our work.

Concerning your questions “Apart from these 5 papers (originating from Italy and the USA), have you encountered this type of research in other countries? Has this topic never been addressed before and never since?”, as we wrote at the lines 76-79, in the paragraph of Methods, our research was carried out using two notorious databases and with no geographical nor temporal limitation. The limited number of article fulfilling the inclusion criteria is related to the paucity of research on this topic worldwide.

In addition, a specification for Your revisions is below. We followed your advice and changed it according to your observation:

  1. "Please note the values of 75 mg/m2 and 100 mg/m2. (Sometimes there are spaces and superscripts, sometimes not.)"

The unity of measure has been corrected as 75 mg/m2 and 100 mg/m2

  1. "There are many spaces in section 2.1".

Thank you for noticing. Spaces have been removed.

  1. "Captions under Table 1 and Table 2 are hard to read (too many spaces)".

Spaces removed.

  1. "In subsection 2.3 it says "FP e DA extracted" and shouldn't it say "FP and DA extracted"?"

Thank you for noticing the mistake, we corrected “and”.

Also, you can find the rewritten and corrected version of the manuscript in the attached file. We highlighted any changes made.

Thank you very much for your advice and comments. We hope we have complied with your requests.

Reviewer 2 Report

Its a very interesting topic,and the revue is salutary.The role of HIPEC is still debatable  and is necessary the analysis of literature becausa it existe a paucity of data about the oncological outcome of association  MIS with HIPEC in advanced EOC.This revue is consistent and emphasyses the clinical advantages and the low rate of complications.The key is the selection of candidates for this therapeuticals combination and to demonstrate some advantages over the open surgery.The results are properly described and the discussions are clear and pragmatic and are presented the limitations of the review.The conclusions are logic and common sense.The references are well chosen and well cited. 

Author Response

Dear Reviewer,

Thank You for taking the time to review our manuscript. They are crucial and valuable to us in raising the quality standard of our work. This is very important to us. We also wanted to inform you that a general revision of English and grammar has been made. 

Thank you for your comment. 

"Its a very interesting topic, and the revue is salutary. The role of HIPEC is still debatable  and is necessary the analysis of literature becausa it existe a paucity of data about the oncological outcome of association  MIS with HIPEC in advanced EOC.This revue is consistent and emphasyses the clinical advantages and the low rate of complications.The key is the selection of candidates for this therapeuticals combination and to demonstrate some advantages over the open surgery.The results are properly described and the discussions are clear and pragmatic and are presented the limitations of the review.The conclusions are logic and common sense.The references are well chosen and well cited. "
